# Multifaceted and Intricate Oncogenic Mechanisms of NDRG1 in Head and Neck Cancer Depend on Its C-Terminal 3R-Motif

**DOI:** 10.3390/cells11091581

**Published:** 2022-05-07

**Authors:** Guo-Rung You, Joseph T. Chang, Hsiao-Fan Li, Ann-Joy Cheng

**Affiliations:** 1Department of Medical Biotechnology and Laboratory Science, College of Medicine, Chang Gung University, Taoyuan 33302, Taiwan; d000017007@cgu.edu.tw; 2Department of Radiation Oncology and Proton Therapy Center, Linkou Chang Gung Memorial Hospital and Chang Gung University, Taoyuan 33302, Taiwan; jtchang@adm.cgmh.org.tw; 3School of Medicine, Chang Gung University, Taoyuan 33302, Taiwan; 4Graduate Institute of Biomedical Sciences, College of Medicine, Chang Gung University, Taoyuan 33302, Taiwan; hflico@gmail.com

**Keywords:** NDRG1, head and neck cancer, cancer stemness, cell motility, 3R-motif, prognosis

## Abstract

N-Myc downstream-regulated 1 (NDRG1) has inconsistent oncogenic functions in various cancers. We surveyed and characterized the role of NDRG1 in head and neck cancer (HNC). Cellular methods included spheroid cell formation, clonogenic survival, cell viability, and Matrigel invasion assays. Molecular techniques included transcriptomic profiling, RT-qPCR, immunoblotting, in vitro phosphorylation, immunofluorescent staining, and confocal microscopy. Prognostic significance was assessed by Kaplan–Meier analysis. NDRG1 participated in diverse oncogenic functions in HNC cells, mainly stress response and cell motility. Notably, NDRG1 contributed to spheroid cell growth, radio-chemoresistance, and upregulation of stemness-related markers (CD44 and Twist1). NDRG1 facilitated cell migration and invasion, and was associated with modulation of the extracellular matrix molecules (fibronectin, vimentin). Characterizing the 3R-motif in NDRG1 revealed its mechanism in the differential regulation of the phenotypes. The 3R-motif displayed minimal effect on cancer stemness but was crucial for cell motility. Phosphorylating the motif by GSK3b at serine residues led to its nuclear translocation to promote motility. Clinical analyses supported the oncogenic function of NDRG1, which was overexpressed in HNC and associated with poor prognosis. The data elucidate the multifaceted and intricate mechanisms of NDRG1 in HNC. NDRG1 may be a prognostic indicator or therapeutic target for refractory HNC.

## 1. Introduction

Head–neck cancer (HNC) is one of the most prevalent cancers worldwide, accounting for approximately 5% of all cancers, according to GLOBOCAN 2020 estimates [1]. The standard treatment methods for HNC include surgery, radiotherapy, and chemotherapy alone or in combination. Although treatment strategies have advanced in the last few decades, the prognosis of patients with HNC has not significantly improved [2]. To investigate the molecular carcinogenesis of HNC, our laboratory previously performed differential display analysis to search for genes with an altered expression between normal and cancer tissues. Among all dysregulated genes, N-myc downstream-regulated gene 1 (NDRG1) was overexpressed in cancer tissues to the greatest extent, suggesting a significant role for this molecule in HNC formation [3]. Cancer metastasis and recurrence after chemoradiotherapy are the significant causes of treatment failure. Interestingly, patients with radioresistant or recurrent cancers often have a higher rate of metastasis [4,5]. Similarly, highly invasive cancers with nodal metastasis are often accompanied by a poor radiotherapeutic response [4,5]. Recently, an association between cell invasion and radioresistance was demonstrated. These two malignant attributes share phenotypic crosstalk, and several molecules with both functions have been identified [6]. To analyze metastatic and radioresistant molecules globally, we employed cDNA microarrays to determine differential transcriptomic profiles. One of the hub genes, N-myc downstream-regulated gene 1 (NDRG1), has been related to cell invasion and radioresistance [6], indicating its potential crucial function in the pathological mechanism of HNC. Nevertheless, the regulation of malignancy in HNC by this molecule has not yet been elucidated.

NDRG1 was first identified in 1996 and is a reducing agent and tunicamycin-responsive protein (RTP) [7]. In 1998, several groups reported the discovery of the gene. One group named the gene *Cap43* (calcium-activated protein, 43 kDa), reflecting its induction after cells are exposed to metal ions [8]. Other groups called this gene *rit42* (reduced in the tumor, 42 kDa) or *Drg1* (differentiation-related gene 1), indicating that the downregulation of this gene is associated with a malignant phenotype [9]. Later, this gene was found to be homologous to two mouse genes, *TDD5* and *Ndr1*, and was widely distributed in a pattern similar to that in other species, such as plants and roundworms [10]. Therefore, *RTP*/*Drg1*/*Cap43*/*rit42*/*TDD5*/*Ndr1* is thought to be evolutionarily conserved in humans, mice, rats, roundworms, and plants, respectively. In humans, this gene is widely expressed in many tissues and is thought to have essential functions [11]. In 2004, the Human Nomenclature Committee (HUGO) officially unified all designations of this gene as NDRG1.

NDRG1 is localized on human chromosome 8q24.3 and is expressed as a 3.0 kb mRNA, including a 1182 bp open reading frame. It is translated into 394 amino acids, comprising a 43 kDa protein [9]. NDRG1 is distributed mainly in the cytoplasm [10] but is also associated with the cellular membrane or nucleus [11,12]. This molecule reportedly participates in diverse cellular functions, including embryonic development, cellular differentiation, genomic stability, metabolism, apoptosis, and autophagy [9,10,11,12,13,14,15,16,17,18,19]. It is also considered a general stress-responsive gene because it can be induced by several stimuli, such as chemicals, metal ions, DNA damage, endoplasmic reticulum stress, or cellular hypoxia [20,21,22,23].

Reports on the role of NDRG1 in modulating tumor development have been inconsistent. NDRG1 may act as an oncogene, as it is overexpressed in many types of cancers, including oral, esophageal, gastric, colon, liver, lung, and bladder cancer [23,24,25,26,27,28,29,30,31,32]. The major oncogenic function of NDRG1 is its role in promoting cellular motility, which enhances cell invasion, migration, or angiogenesis [26,27,28,29,30,31,32,33]. Other oncogenic functions of NDRG1 have been reported, including facilitation of tumorigenesis (cell growth/tumor initiation/stemness conversion) [34,35,36,37] and increasing radio-chemoresistance [19,38,39,40,41,42]. Paradoxically, NDRG1 is also a putative tumor suppressor as it is downregulated in several types of cancers, such as prostate, renal, pancreatic, and endometrial cancer, and neuroblastoma [43,44,45,46]. Several tumor-suppressive functions of NDRG1 have been reported, including the reduction in cellular motility (invasion/migration) [44,45,46,47,48,49,50,51], suppression of tumorigenesis (cell growth/stemness conversion) [43,50,51,52], and induction of apoptosis [16,17]. Therefore, NDRG1 may play diverse roles in modulating malignant formation that may depend on a specific tissue type or may have differential roles under certain cellular conditions. However, the function of NDRG1 in HNC has not yet been addressed.

In this study, we surveyed the molecular roles of NDRG1 via transcriptomic analysis in HNC cells. The data reveal that it mainly regulates cancer stemness and cell motility. We further characterized an NDRG1-specific domain, the 3R motif, and discovered a deterministic mechanism in the differential regulation of these phenotypes. We defined the clinical role of NDRG1 as an oncogene because it is overexpressed in tumor tissues and is associated with poor prognosis in patients with HNC. These data elucidate the intricate mechanisms of NDRG1 in cancer cells and implicate this molecule as a prognostic indicator for HNC.

## 2. Materials and Methods

### 2.1. Cell Lines and Cellular Transfection

OECM1, SAS, and FaDu HNC cell lines were used. These cell lines were easy to maintain and efficient in plasmid transfection. The cell culture conditions were the same as previously described [53,54,55]. OECM1 cells were cultured in RPMI-1640 medium, SAS cells in Dulbecco’s modified Eagle medium (DMEM), and FaDu cells in MEM. Cellular transfection was performed as previously described [53,54,55]. Briefly, cells were transfected with an NDRG1-specific plasmid or the vector plasmid using Lipofectamine 2000 (Invitrogen, Carlsbad, CA, USA), followed by incubation in Opti-MEM (Invitrogen, Carlsbad, CA, USA) for 12 h. The cells were continuously cultured in a complete medium. To clone stably transfected cells, the neomycin reagent, a G418 antibiotic solution (Sigma-Aldrich, St. Louis, MO, USA), was used to select stably transfected cell lines.

### 2.2. Construction of Full-Length and Truncated NDRG1 Expression Plasmids

To construct the full-length plasmid of NDRG1 (NDRG1-FL), the NDRG1 open reading frame was prepared from cDNA from the OECM1 cancer cell line by PCR (forward: 5′-atg-tct-cgg-gag-atg-cag-ga-3′; reverse: 5′-cta-gca-gga-gac-ctc-cat-g-3′) and cloned into the pGEM-T Easy vector (Promega, Madison, WI, USA). The NDRG1 cDNA sequence was subcloned into the pEGFP-C1 vector, which contained an in-frame sequence for enhanced green fluorescent protein (EGFP). An expression plasmid containing the NDRG1 deleted C-terminal domain (NDRG1-dC) was prepared from the NDRG1-FL plasmid by PCR (forward: 5′-ccc-aag-ctt-atg-tct-cgg-gag-atg-cag-gat-3′; reverse: 5′-cg-gga-tcc-cag-aga-agt-gac-gct-gga-acc-a-3′). The PCR product was digested with HindIII and BamHI, and subcloned into the pEGFP-C1 vector. An expression plasmid for NDRG1 with three point mutations (S342A, S352A, and S362A) in its C-terminal region (NDRG1-3mt) was prepared from the NDRG1-FL plasmid via point mutagenesis and subcloned into the pEGFP-C1 vector.

The NDRG1-3mt plasmid with triple point mutations of NDRG1 were synthesized using QuickChange site-directed mutagenesis kit (Stratagene, La Jolla, CA, USA). We applied three times of point mutagenesis experiments to generate a triple mutation construct in the S342A, S352A, and S362A. Briefly, this mutant construct was generated from NDRG1-FL plasmid by standard PCR reaction using the wild-type dsDNA as a template and three pairs of mutant primers (S342A forward: 5′-gatggcacccgcgctcgctcccaca-3′, reverse: 5′-gatggcacccgcagccgctcccaca-3′; S352A forward: 5′-agggcacccgagctcgctcccacac-3′, reverse: 5′-gtgtgggagcgagctcgggtgccct-3′; S362A forward:5′-agggcacccgcgctcgctcgcacac-3′, reverse: 5′-gtgtgcgagcgagcgcgggtgccct-3′). After PCR amplification by PfuUltra HF DNA polymerase, the Dpn I restriction enzyme was added to digest the wild-type dsDNA template. Finally, the Dpn I-treated DNA was transformed into competent cells to produce NDRG1-3mt plasmid.

### 2.3. Transcriptomic and Bioinformatic Analyses in a Global Survey of Molecular Functions

After transfection of the full-length NDRG1 (NDRG1-FL) or the vector plasmids into OECM1 cells, the differential transcriptomes between NDRG-FL and the vector were determined using Exon array analysis. The biotinylated oligonucleotide was hybridized to Affymetrix Gene Chip ™ Human Exon 1.0 ST array chip (Affymetrix, Santa Clara, CA, USA) and scanned by an Affymetrix Gene Array 2500 scanner (Affymetrix, Santa Clara, CA, USA). The microarray raw signal intensities were imported into Partek^®^ Genomics Suite v6.3 (Partek, St. Louis, MO, USA) and normalized using the robust multiarray averaging (RMA) algorithm. After scaling and applying threshold and ceiling values (floor 20, ceiling 20,000), the RMA normalized data were analyzed using the analysis of variance (ANOVA) to determine samples based on the two-fold change. The two-fold change was set as the threshold for screening differentially expressed genes (DEGs) by comparing the vector control cells to the NDRG1-overexpressed cells using the analysis of variance (ANOVA).

To globally survey the differential transcriptome, we used the two-over-representation analysis, such as GO and KEGG, to identify the specific functional mechanism of NDRG1 based on the downstream network molecules and specific pathways. Furthermore, to determine significant, concordant differences between two biological phenotypes, we performed the GSEA analysis to uncover what specific phenotypes were predominant in NDRG1 overexpressed in the OECM1 cells by the annotated gene sets. To study the function of molecules and pathways, the DEGs list was uploaded to the Database for Annotation, Visualization, and Integrated Discovery (DAVID) website (https://david-d.ncifcrf.gov/, accessed on 14 February 2021), including Gene Ontology (GO) and Kyoto Encyclopedia of Genes and Genomes (KEGG) pathway analyses. Pie plots of functional categories were generated using GO analysis. According to the KEGG database, functional pathways were identified for enrichment bubble plots of the functional pathway. Functional pathways were identified according to the KEGG database for enrichment bubble plots of the functional pathway [53]. For gene set enrichment analysis (GSEA; http://software.broadinstitute.org/gsea/index.jsp, accessed on 6 May 2021), the DEGs profile was used as a gene set for enrichment analysis to identify a statistically significant phenotype. The samples were first divided into vector control groups and NDRG1-overexpressed groups according to the transfection conditions. Then, the C2 curated gene set (c2.all.v7.4.symbols.GMT) was downloaded from the Molecular Signatures Database (MSigDB). This collection was used as a reference gene set for GSEA software.

### 2.4. Assessment of Cancer Stemness by Spheroid Cell Formation

Cancer stemness was assessed based on the ability to form spheroids as previously described [54,55,56]. Briefly, the cells were diluted as a suspension (1000 cells/mL) in a culture medium containing DMEM/F12 medium supplemented with 10 ng/mL epidermal growth factor (EGF; Invitrogen, Carlsbad, CA, USA) and 10 ng/mL basic fibroblast growth factor (bFGF; Invitrogen, Carlsbad, CA, USA), N-2 supplement (Invitrogen, Carlsbad, CA, USA), B-27 supplement (Invitrogen, Carlsbad, CA, USA), and 1% Matrigel (BD Biosciences, San Jose, CA, USA). Subsequently, 1 mL of the cell suspension was plated in each well of a Costar^®^ 24-well ultra-low attachment plate (CORNING, Corning, NY, USA). After 10 days, cell spheres were visualized and enumerated.

### 2.5. Determination of Radiosensitivity and Chemosensitivity

Radiosensitivity and chemosensitivity were determined using clonogenic survival and viability assays based on MTS, as previously described [54,55,56]. The MTS assay was performed to assess chemosensitivity using the CellTiter 96 Aqueous One Solution Cell Proliferation Assay kit (Promega, Madison, WI, USA). Briefly, the cells seeded in the 96-well plate were treated with serial concentrations of cisplatin. After 48 h of incubation, MTS reagent was added to each well, and the cell viability was determined by the absorbance at 490 nm under a multimode microplate reader (Molecular Devices, San Jose, CA, USA). To determine radiosensitivity, cells were seeded in a 6-well cell culture plate for 8 h. The cells were then exposed to radiation (0–6 Gy) and continuously cultured for 7–14 days to allow cell colony formation. The cell colonies were stained with crystal violet, and the survival fraction was calculated as the number of colonies divided by the number of seeded cells multiplied by plating efficiency. To measure chemosensitivity, the cells were treated with various concentrations (0–5 μg/mL) of cisplatin and maintained in the culture medium for two days. The number of surviving cells was counted and compared with that of untreated surviving cells.

### 2.6. Analyses of Cell Invasion and Migration

Cell invasion was determined using a Matrigel invasion assay as previously described [54,55,56]. Briefly, the membrane in the upper chamber of a Millicell apparatus (Millipore, Burlington, MA, USA) was coated with Matrigel (BD Biosciences, San Jose, CA, USA). Cells in a medium containing 1% fetal bovine serum (FBS) were seeded into the upper chamber, while the lower chamber contained 20% FBS in the medium to attract invading cells. After incubation, the cells that had passed through the membrane to invade the reverse side were stained and photographed. Cell migration was examined using an in vitro wound healing assay as previously described [54,55,56]. Briefly, the cells were seeded in an ibidi^®^ culture insert (Applied BioPhysics, Inc., Troy, NY, USA) on top of a 6-well plate. After 12 h of incubation, the culture insert was detached to form a cell-free gap in the monolayer. After changing to a culture medium containing 1% FBS, cell migration into the gap was photographed every 8 h.

### 2.7. RNA Extraction and RT-qPCR Analysis

RNA extraction and cDNA synthesis were performed as previously described [56,57]. Briefly, PCR and cDNA synthesis were performed using a MiniOpticon™ real-time PCR detection system with SYBR Green Supermix reagents (Bio-Rad Lab., Hercules, CA, USA). The qPCR amplification reactions were carried out with 35 cycles of denaturation at 95 °C for 15 s, annealing at 55 to 60 °C for 30 s, and extension at 72 °C for 1 min. Primers used in this study are listed in Appendix A.

### 2.8. Protein Extraction, Subcellular Fractionation, and Western Blot Analysis

Protein extraction and Western blot analyses were performed as previously described [57,58]. Briefly, cellular proteins were extracted or fractionated subcellularly via the NE-PER nuclear and cytoplasmic extraction reagent (Thermo Fisher Scientific, Waltham, MA, USA). Proteins were separated by SDS-PAGE and transferred onto nitrocellulose membranes. After hybridization with primary antibodies and a secondary antibody conjugated with horseradish peroxidase, the membranes were developed using an ECL reagent (EMD Millipore, Darmstadt, Germany), followed by autoradiography. Glyceraldehyde 3-phosphate dehydrogenase (GAPDH) level was used as an internal control. The relative density of each sample was determined after normalization to GAPDH levels. All the antibodies used in this study are listed in Appendix A.

### 2.9. Immunofluorescence Analysis and Confocal Microscopic Examination

Immunofluorescence and confocal microscopy were performed as previously described [56,57,58]. Briefly, EGFP-fused NDRG1-expressing cells were grown on coverslips coated with poly-l-lysine, fixed in paraformaldehyde, permeabilized, and blocked with FBS. After fixation, the cells were mounted with a mounting medium containing DAPI fluorescent dye (Vectashield H-1200; Vector Lab, Burlingame, CA, USA). Immunofluorescence was visualized using an LSM 510 META confocal microscope (Zeiss Microscopy GmbH, Jena, Germany).

### 2.10. Immunoprecipitation, Kinase Assay, and In Vitro Assessment of Phosphorylation

Glycogen synthase kinase 3-beta (GSK3b) kinase inhibitor (SB216763) was purchased from Sigma-Aldrich. To determine the phosphorylation status of NDRG1, immunoprecipitation was used to pull down the NDRG1 protein, followed by an in vitro phosphorylation assay, similarly as previously described [57,58,59]. Briefly, Protein A/G PLUS-agarose beads (Santa Cruz Biotechnology, Dallas, TX, USA) with an anti-GFP antibody were used to precipitate NDRG1 proteins. After elution of the immunoprecipitated complex, the protein solution was subjected to an in vitro kinase assay by incubation with recombinant GSK3b protein (Sigma-Aldrich, St. Louis, MO, USA) in a solution containing ATP. Phosphorylated proteins were subjected to SDS-PAGE analysis, and the phosphorylation status of NDRG1 was determined by hybridization with an anti-phosphoserine antibody (Abcam, Cambridge, UK). Alternatively, NDRG1-FL- or NDRG1-Dc-transfected cells were treated with GSK3b-specific inhibitor (SB21673) at various doses for 24 h. Cellular proteins were extracted and subjected to a pulldown assay with an anti-GFP antibody. The phosphorylation status of NDRG1 was revealed by phospho-serine-specific immunoblotting.

### 2.11. Clinical Tissue Expression and Prognostic Evaluation of NDRG1 in Patients with HNSC

The Cancer Genome Atlas (TCGA)-HNSC cohort was used for the clinical analysis. The GEPIA2 platform (http://gepia2.cancer-pku.cn/#index, accessed on 4 May 2021) was used to compare the transcript levels of NDRG1 in HNSC tumors (N = 519) and normal tissues (N = 44). The KM-Plotter online tool (http://kmplot.com/analysis, accessed on 4 May 2021) was used to determine the prognostic significance of NDRG1 in patients with HNSC (N = 500). High- and low-risk groups were classified using an optimization algorithm in the order of the prognostic index, according to the gene expression level. Kaplan–Meier analysis was performed to evaluate the overall survival at the five-year (60 months) follow-up. The hazard ratios (HRs) with 95% confidence intervals (CIs) and log-rank *p*-values were calculated using the Kaplan–Meier method and Cox regression analysis.

## 3. Results

### 3.1. Transcriptomic Analysis Reveals NDRG1 Participation in Diverse Oncogenic Functions

To survey the molecular functions of NDRG1 globally, transcriptomic and functional pathway analyses were performed using microarray and bioinformatics methods. After transfection of the full-length NDRG1 (NDRG1-FL) plasmid into HNC cells (OECM1, SAS, and Fadu), the NDRG1 protein expression was determined by Western blot analysis. As shown in Figure 1a, NDRG1 was substantially increased in all cell lines, indicating a successful transfection of these plasmids. Furthermore, NDRG1 overexpressed cells exhibited a mesenchymal-like, spindle, or fibroblast morphological alteration compared to the parental OECM1 cells with the epithelial-like morphology (Figure 1b). This result implied that the mechanism of epithelial-to-mesenchymal transition may be partially involved.

The differential transcriptomes were determined using Affymetrix Exon array analysis in OECM1 cells. The exon array data of NDRG1-FL were compared with those of vector transfectants by a two-fold change in expression by ANOVA analysis. A total of 1109 DEGs were identified, including 475 upregulated and 634 downregulated genes. The top 50 upregulated and 50 downregulated genes are shown in Appendix A.

These genes were subjected to DAVID analysis for gene annotation and network pathway analysis. NDRG1 participated in a wide range of cellular functions, including the growth (20%), migration/invasion (12%), stress response (10%), signal transduction (10%), and development/differentiation (8%) (Figure 1b). GO analysis revealed the role of NDRG1 in stress response mechanisms that included DNA replication and repair, cellular apoptosis, drug resistance, molecular signaling, and stemness (Figure 1c). GSEA analysis confirmed that NDRG1 was significantly enriched in these malignant-related phenotypes, such as regulation of stemness (*p* < 0.001) and epithelial-to-mesenchymal transition (*p* < 0.001) (Figure 1d). These results suggested that NDRG1 participates in a wide range of oncogenic functions.

### 3.2. NDRG1 Contributes to Cancer Stemness and Leads to Chemo-Radioresistance

Cancer stemness is characterized by stress tolerance and resistance [60,61]. The transcriptomic analysis data indicated that NDRG1 fulfills the attributes of cancer stemness. To confirm the potential effect of this molecule, we transfected the NDRG1-FL plasmid into HNC cells and determined the specific cancer stemness ability presented by spheroid cell formation. As shown in Figure 2a, overexpression of NDRG1 significantly promoted cancer stemness in the two HNC cell lines, as observed by the increase in the number and size of the cellular spheres (*p* < 0.001 in OECM1 cells and *p* < 0.01 in FaDu cells).

Since radioresistance and chemoresistance are concurrent presentations of cancer stemness [60,61], we examined whether NDRG1 also regulates these functions. A clonogenic survival assay was performed to assess the effect of radiosensitivity. As shown in Figure 2b, NDRG1 overexpression led to higher resistance to irradiation in two tested cell lines (*p* < 0.001 at 6 Gy in both OECM1 and FaDu cell lines). The MTS survival assay was performed to determine the effect of cisplatin treatment on chemosensitivity. As shown in Figure 2c, NDRG1 overexpression resulted in higher drug resistance (*p* < 0.001 at 3 μg/mL in OECM1 cells and 5 μg/mL in FaDu cells).

To extend the cellular effect of NDG1 to molecular presentations, we determined the expression of cancer stemness-associated regulators (ABCG2, Twist1, BMI1, NES, c-Met) in NDRG1-FL-transfected HNC cells. RT-qPCR was performed to examine mRNA expression status. Although there were various levels in the two HNC cell lines, these molecules were generally elevated in response to NDRG1 transduction (Figure 2d). The protein levels of CD44 and Twist1 were confirmed by Western blot analysis, which showed significant elevation in response to NDRG1 transduction (Figure 2e). The collective findings indicated that NDRG1 contributes to cancer stemness and chemo-radioresistance, which may be related to several stemness regulators.

### 3.3. NDRG1 Facilitates Cell Motility via Modulation of Extracellular Matrix (ECM)

To examine the potential effect of NDRG1 on cell motility, in vitro wound healing and Matrigel invasion assays were performed. As shown in Figure 3a, transfection with NDRG1-FL promoted cell migration, with a 1.8–2.7-fold increase observed in the three HNC cell lines. Similarly, overexpression of NDRG1 facilitated cell invasion, with a 1.3–2.5-fold increase observed in all HNC cell lines (Figure 3b). Conversely, silencing NDRG1 inhibited cell migration and invasion in OECM1 cells (Appendix A).

To extend the molecular presentation, we examined the expression of ECM-associated molecules, including fibronectin, vimentin, and N-cadherin. The RT-qPCR and Western blot methods were used to determine the expression levels of mRNA and proteins. These molecules were significantly upregulated at various levels in response to NDRG1 transduction in the two tested cell lines at both the mRNA (Figure 3c) and protein levels (Figure 3d). These results suggest that NDRG1 promotes cell motility in HNC cells via modulation of the ECM-related pathways.

### 3.4. 3R-Motif of NDRG1 Has Minimal Effect on Cancer Stemness

NDRG1 belongs to the NDRG gene family, which comprises an α/β hydrolase fold region. This region has been widely examined by several investigators previously. The C-terminal region of the NDRG1 protein contains a unique three-tandem-repeat (3R) motif comprised of a 10 amino acid sequence (GTRSRSHTSE) [11]. However, the molecular function of the 3R motif has not yet been examined. Thus, we characterized this unique structure and delineated its role in malignant-related phenotypes. To examine whether this motif plays a biological role, we constructed a plasmid expressing an NDRG1 isoform with a C-terminal deletion (NDRG1-dC) (Figure 4a). After transfection and expression confirmation of the protein levels by the Western blot method in the two HNC cell lines (Figure 4b), the effects of C-terminal deletion on the functions of cancer stemness and cell motility were examined.

The effects on cancer stemness are shown in Figure 4c. Spheroid cell formation was significantly increased in response to NDRG1-FL transfection. This ability was also observed in NDRG1-dC transfectants, suggesting that the 3R-motif of NDRG1 has a negligible effect on cancer stemness. Similar results were observed for NDRG1 in radioresistance. As shown in Figure 4d, transduction with NDRG1-FL promoted radioresistance in OECM1 cells. This effect existed even without the C-terminal domain, indicating a minimal effect of the 3R-motif on cancer stemness.

The molecular regulation of cancer stemness-related biomarkers/regulators (BMI1, NES, and c-Met) was also examined in OECM1 cells. As shown in Figure 4e, transduction with NDRG1-dC, similar to that with NDRG1-FL, substantially induced the expression of these molecules. These molecular results support the cellular investigations that NDRG1 contributes to cancer stemness with a minor role by its 3R-motif.

### 3.5. 3R-Motif of NDRG1 Is Critical for Cell Motility

The effect of the 3R-motif on cell motility was examined by transduction of NDRG1-dC into HNC cells. In the cell migration assay, migration was significantly increased upon NDRG1-FL transfection in all three tested cell lines (Figure 5a). However, this increase was diminished upon deprivation of the 3R-motif in NDRG1-dC transfectants. Similar results for cell invasion were observed (Figure 5b). Cell invasion was promoted by NDRG1-FL but was abolished in NDRG1-dC transfectants. These cellular results were supported by molecular presentations (Figure 5c). The ECM-related molecules fibronectin, vimentin, were significantly upregulated by NDRG1-FL but were not induced by NDRG1-dC (Figure 5c). These results suggest that NDRG1 facilitates cell motility, and that its 3R motif plays a crucial role in this effect.

### 3.6. 3R-Motif of NDRG1 Is Required for Nuclear Translocation and Phosphorylation

The 3R-motif of NDRG1 (NDRG1-3R) has not yet been characterized. Therefore, we searched for a homologous protein sequence to this domain using the SeqWeb searching tool. This 3R-motif was similar in sequence to the RS domains (arginine/serine-rich domain) of the SR protein family (Figure 6a). Because SR proteins are reportedly involved in cellular trafficking and subcellular translocation [62,63,64], we examined whether NDRG1-3R also plays a role in guiding nuclear translocation. Immunofluorescence staining and confocal microscopy were used to determine the specific subcellular localization of NDRG1 in HNC cells. Green staining indicated a particular site of NDRG1 (EGFP-tag), while blue staining with DAPI dye revealed nuclei. Similar results were observed for the three HNC cell lines. In the vector- or NDRG1-FL-transfected cells, this protein was distributed more evenly in all cellular compartments, including the cytoplasm and nucleus (Figure 6b). However, NDRG1 was not detected in the nuclei of cells in NDRG1-dC transfectants. Apparently, deprivation of the 3R-motif prohibited NDRG1 import into the nucleus.

To confirm the guiding function of this 3R-motif, Western blot analysis of subcellular protein fractions was performed. The expression levels of HDAC1 and GAPDH were used as internal controls to indicate nuclear and cytosolic compartments. Two HNC cell lines were examined, and the results were consistent (Figure 6c). Both NDRG1-FL and NDRG1-dC proteins were observed in the cytosolic fraction, whereas only NDRG1-FL was found in the nucleus. NDRG1 truncated with deprivation of the 3R-motif (NDRG1-dC) was excluded from the nuclear fraction. These results suggested that NDRG1-3R plays an essential role in guiding nuclear translocation.

As the kinase at the RS domain of SR proteins is essential for nuclear import [62,63,64], we determined whether the phosphorylation of NDRG1-3R is critical for its nuclear import. Because this 3R-motif possesses three putative sites (Ser342, Ser352, and Ser362) for phosphorylation by GSK3b [65], we examined the effect of this kinase on NDRG1 in OECM1 cells. Cellular proteins extracted from NDRG1-FL or NDRG1-dC transfectants were analyzed by an in vitro phosphorylation assay with recombinant GSK3b protein. Phosphorylation status was revealed by immunoblotting with a phosphoserine-specific antibody after being pulled down with anti-GFP antibodies. Phosphorylation of NDRG1-FL substantially increased in response to GSK3b treatment (Figure 6d). However, NDRG1-dC protein was insensitive to this kinase and exhibited minimal phosphorylation. These results indicated the specific kinase activity of GSK3b for NDRG1 at the 3R-motif residues.

We then examined whether NDRG1 phosphorylation by GSK3b was critical for its nuclear import. OECM1 cells were treated with a GSK3b-specific inhibitor (SB21673), followed by protein extraction by subcellular fractionation and analysis of NDRG1 expression by Western blotting analysis. As shown in Figure 6e, treatment with the GSK3b inhibitor substantially reduced NDRG1 protein levels in the nuclear compartment, whereas no difference was observed in the cytoplasm. The effect of GSK3b on NDRG1 translocation was confirmed by confocal microscopy. Two HNC cell lines were tested, and similar results were obtained. As shown in Figure 6f, NDRG1 was evenly distributed in the nucleus and cytoplasm of control cells. Inhibition of NDRG1 phosphorylation by a GSK3b inhibitor attenuated its presentation in the nuclei. Thus, GSK3b phosphorylation at the 3R-motif of NDRG1 may be indispensable for its molecular import to the nucleus.

### 3.7. Phosphorylation of the 3R-Motif by GSK3b Is Critical for Cell Motility in HNC Cells

To investigate whether GSK3b kinase activity at the 3R-motif is essential for cell motility, we examined the effect of cell invasion after suppressing this kinase ability using a GSK3b-specific inhibitor (SB21673). Two cell lines were examined, and similar results were obtained. Transfection with NDRG1-FL significantly increased cell invasion (Figure 7a). However, the inhibition of GSK3b kinase abrogated NDRG1-induced cell invasion.

To further validate that the phosphorylation sites at the 3R-motif are essential for cell invasion, we constructed an NDRG1-expressing plasmid with three serine point mutations at the 3R-motif (Ser342, Ser352, and Ser362) (NDRG1-3mt). The differential invasion ability of HNC cells after transfection with NDRG1-FL, -dC, or -3mt in HNC cells were examined. As shown in Figure 7b, NDRG1-FL transfection significantly increased cell invasion. However, this function was minimized in transfectants by either NDRG1-dC or NDRG1-3mt. Therefore, the cell motility function promoted by NDRG1 is dependent on its 3R-motif. This motif must be phosphorylated by GSK3b to guide its nuclear translocation.

### 3.8. NDRG1 Is Overexpressed in Tumors and Is Associated with Poor Prognosis in Patients with HNC

To determine the definitive role of NDRG1 in HNC, we determined the clinical presentation of this molecule in patients with HNC. Differential expression of NDRG1 between normal and tumor tissues was examined using The Cancer Genome Atlas Head-Neck Squamous Cell Carcinoma (TCGA-HNSC) dataset. As shown in Figure 8a, NDRG1 was overexpressed in tumors (*p* < 0.001), supporting the oncogenic function of this molecule in cancer formation. We further assessed the association between NDRG1 expression and patient survival using the TCGA-HNSC cohort (N = 500). As shown in Figure 8b, high NDRG1 levels were significantly correlated with poor survival (*p* = 0.034, HR = 1.37). These results suggested that NDRG1 functions in cancer aggressiveness. NDRG1 may be a prognostic biomarker and therapeutic target for patients with HNC.

## 4. Discussion

NDRG1 is a multifunctional protein that participates in several cellular processes to maintain homeostasis. Although the role of NDRG1 in various types of cancer remains elusive, the present findings reveal its oncogenic function in HNC. Transcriptomic profiling has revealed that NDRG1 participates in a wide range of oncogenic functions. Bioinformatic analysis revealed the role of NDRG1 in the stress response, including DNA repair, cellular apoptosis, drug resistance, and stemness (Figure 1c). We further validated these roles by showing that NDRG1 promotes cancer stemness conversion and contributes to chemo-radioresistance in HNC cells (Figure 2). These results have been supported by other investigations [34,35,36,37,38,39,40,41]. A high level of NDRG1 was associated with the resistance to ionizing radiation [6,38] or multiple drug-induced cytotoxicities [15,39,40,41] in various types of cancer cells. Conversely, knockdown of this molecule sensitized cells to irradiation or response to drug treatment [15,38]. Moreover, a clinical study revealed that higher NDRG1 levels correlated to a worse therapeutic outcome of neoadjuvant chemotherapy in the patients with esophageal squamous carcinoma [42]. Furthermore, our transcriptomic analysis revealed that NDRG1 regulates cell motility, including the mechanism of epithelial-to-mesenchymal transition (Figure 1d). We also confirmed this cellular function by demonstrating that NDRG1 facilitates cell migration and invasion by modulating ECM (Figure 3). This function of NDRG1 in augmenting cancer metastasis is also supported by other investigations [26,27,28,29,30,31,32,33]. In mice model study, the effect of NDRG1 in tumorigenesis showed enhancement [25,33,34] or suppression [43,50,52] in various studies. However, NDRG1 promoting cancer metastasis is consistently found by several investigators [25,33]. Thus, NDRG1 facilitating cancer metastasis may be the primary cause of cancer progression.

NDRG1 belongs to the NDRG gene family, which comprises an α/β hydrolase fold region (Figure 6a) [66,67,68,69]. With its high serine and threonine content within the α/β hydrolase region, NDRG1 was known to possess potential phosphorylation sites for PKC, PKA, CK-II, and tyrosine kinase [6,68,69]. The PKA and PKC have been shown to phosphorylate NDRG1, which affected its function in cell growth [10,68,69,70]. A truncated form of NDRG1 with a deletion in the N-terminal domain was found in several prostate and pancreatic cancer cell lines. This truncated isoform was located more in the cytosol than the full-length molecule [71]. However, the C-terminal region of NDRG1 with the 3R-motif is a unique structure absent in other NDRG family members [11]. In this study, we discovered that this 3R-motif accounts for differential malignant functions, with minimal effect on cancer stemness (Figure 4), and is indispensable for cell motility (Figure 5). Further investigation of NDRG1-3R revealed that this 3R-motif acts as a leading sequence that can direct NDRG1 translocation into the nucleus (Figure 6b,c). This result may explain previously obscure descriptions of NDRG1 localization to the nuclear compartment in the absence of a defined localization sequence [7,12]. Our novel findings of the 3R-motif from nuclear import to cell motility resolved the clinical presentation. In colorectal cancer patients with lymph node metastases, the nuclear localization of NDRG1 was significantly higher in the neoplasm tissue than in the normal mucosa. However, there was no differential localization in the cytosolic compartment [27]. In gastric cancer, NDRG1 nuclear localization is frequently found in the region of cancerous invasion and is correlated with lymph node metastasis [28]. In prostate cancer, a phosphorylated form of NDRG1(S330) predominantly located in the nucleus was significantly overexpressed in the metastatic lesions of the tumor tissue [72]. Therefore, the function of NDRG1 in promoting cell motility in HNC may rely on its 3R-motif to guide nuclear translocation.

NDRG1 contains multiple putative phosphorylation sites throughout its protein sequence. Several kinase sites for PKC and PKA within the α/β hydrolase region affect NDRG1 functions in growth control [10,68,69,70]. In the C-terminal region of NDRG1, five sites for SGK1 kinase (Thr328, Ser330, Thr346, Thr356, and Thr366) and three sites for GSK3b kinase located at the 3R-motif (Ser342, Ser352, and Ser362) have been predicted [65]. Phosphorylation by SGK1 has been related to several physiological conditions, primarily stress and growth arrest [73,74,75,76,77]. The phosphorylation of NDRG1 by SGK1 at Ser330, but not Thr346, was associated with nuclear localization, and the cytosolic form of the pNDRG1-pT346 was predominant in prostate cancer cells [71]. Conversely, a recent study showed that the phosphorylation of NDRG1 by PIM1 at Ser330 reduced its nuclear localization, and this pNDRG1(S330) was associated with a higher grade of prostate cancer [78]. It was assumed that the differential location of NDRG1 may possess various cellular functions, as NDRG1 translocation may respond to multiple cellular environmental stress conditions. For example, treatment of iron chelators or exposure cells under hypoxia conditions may include NDRG1 upregulation and nuclear localization, along with growth suppression or DNA repairing [11,71]. Thus, differential localization of NDRG1 may be essential to fulfill its multiple functions that may be intricately regulated by various phosphorylation pathways in tissue-specific manners

The effect of GSK3b kinase has not yet been investigated. Previously, NDRG1 has been reported interacting with GSK3b to promote tumor growth [79]. The present study provides the first evidence that GSK3b specifically phosphorylates NDRG1 at the serine residues in the 3R-motif. Truncated NDRG1-dC did not respond to GSK3b kinase activity, whereas it was observed in the NDRG1-FL protein (Figure 6d). Conversely, treatment with a GSK3b-specific inhibitor substantially reduced NDRG1 phosphorylation (Figure 6e). The functional relevance of GSK3b was revealed. Dephosphorylation of NDRG1 by a GSK3b inhibitor prohibited its nuclear import, suggesting that NDRG1 phosphorylation by this kinase is crucial for its nuclear translocation (Figure 6f). Furthermore, we showed that the nuclear import of NDRG1 is closely linked to its function in cellular motility. Dephosphorylation of NDRG1 by GSK3b inhibitor reversed the effect of NDRG1-induced cellular invasion (Figure 7a). The functional significance of GSK3b-mediated phosphorylation of specific sites in NDRG1 was confirmed. Mutation of GSK3b phosphorylation sites at serine residues in the 3R motif reversed NDRG1-induced cellular invasion (Figure 7b). Thus, our results reveal the significance of the 3R-motif in NDRG1. The phosphorylation of NDRG1 by GSK3b in the 3R-motif is critical for NDRG1 nuclear translocation, which is indispensable for the role of NDRG1 in promoting cell motility.

Previously, the confounding role of NDRG1 was reported in HNC. In favor of a tumor-suppressive function, NDRG1 was found to inhibit cell proliferation, invasion, and tumorigenesis in oral cancer cells [50]. A study in the tissue section showed that a lower level of NDRG1 was related to metastatic tumors; however, the expression level of NDRG1 was correlated with invasion markers MMPs [80]. In favor of an oncogenic function, upregulated NDRG1 was associated with high invasion and radioresistance in HNC cells [6]. In two investigations of global molecules using paired clinical oral tissues, NDRG1 was consistently overexpressed in the tumors compared to the normal counterparts [3,81]. These inconsistent findings imply that NDRG1 may possess diverse roles and function differentially under specific cellular conditions. Nevertheless, the clinical presentation of a molecule defines its pathological role. In analyzing the TCGA dataset with 519 HNSC patients, we found that NDRG1 was overexpressed in tumor tissues (Figure 8a), indicating this molecule functions in modulating cancer development. Furthermore, high levels of NDRG1 correlated with poor prognosis (Figure 8b). This result may be caused by the malignant attributes of NDRG1 in promoting cancer stemness (Figure 2) and cell motility (Figure 3). Two molecular mechanisms presumably manifest the aggressive phenotype of NDRG1. Cytoplasmic NDRG1 confers cancer stemness and therapeutic resistance (Figure 4 and Figure 6). NDRG1 may also translocate into the nucleus via phosphorylation at its 3R-motif by GSK3b to facilitate motility (Figure 5, Figure 6 and Figure 7). Thus, our data elucidate the intricate oncogenic mechanisms of NDRG1 in HNC.

## 5. Conclusions

This study reveals the comprehensive role of NDRG1 in HNC. Clinically, NDRG1 is overexpressed in tumors and correlated to poor prognosis. Functionally, this molecule contributes to cancer stemness, therapeutic resistance, and cell motility. Mechanically, various cellular compartments of NDRG1 play differential oncogenic roles. The cytoplasmic NDRG1 may exert its effects on cancer stemness and therapeutic resistance. NDRG1 may also translocate into the nucleus via phosphorylation at its 3R-motif to facilitate motility. The data elucidate the multifaceted and intricate functions of NDRG1 in HNC. NDRG1 may be a prognostic indicator or therapeutic target for refractory HNC.

## Figures and Tables

**Figure 1 cells-11-01581-f001:**
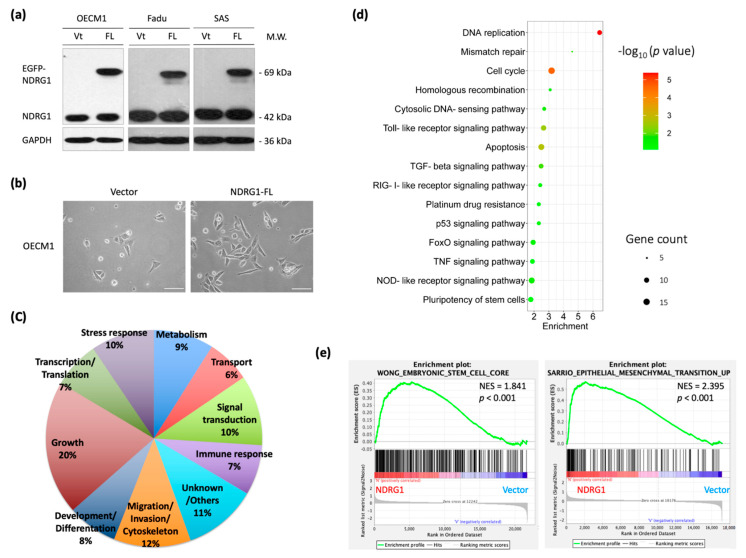
Global survey of molecular mechanisms reveals NDRG1 functions mainly in the regulations of stemness property and invasion phenotype. (**a**) Successful transfection of NDRG1-FL in three HNC cell lines. After transfection of NDRG1 overexpression (FL) or the vector control (Vt) plasmid into HNC cell lines (OECM1, SAS, and FaDu), the NDRG1 protein expression was determined by Western blot analysis. Because the EGFP protein sequence was fused in-frame in the NDRG1 expressing plasmids, the NDRG1 protein in Western blot gel appeared by two bands: the lower one of the endogenous cellular NDRG1 and the higher one of the transfected NDRG1 protein. The GAPDH level was used as an internal control. (**b**) The morphology of OECM1 cells transfected with vector or the NDRG1-FL plasmids, scale bar 20 µm. (**c**) The transcriptomic and bioinformatic analyses for the overexpression of NDRG1 in OECM1 cells. Affymetrix Exon array was performed, and the differential expressed genes (DEGs) were identified by ANOVA analysis with the two-fold change in the gene expression level. Distribution of NDRG1 regulated downstream molecules with functional category, as determined by GO analysis from DAVID website. (**d**) The most significant KEGG functional pathways are regulated by NDRG1, as determined by KEGG enrichment analysis from the DAVID website. (**e**) GSEA analysis of the stemness property and invasion phenotype in vector control or NDRG1 overexpressed in the OECM1 cells.

**Figure 2 cells-11-01581-f002:**
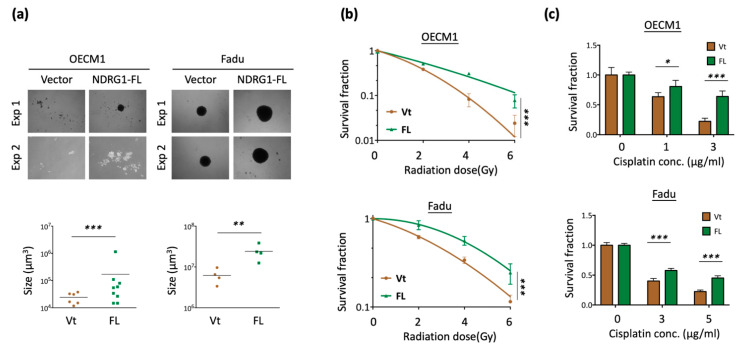
NDRG1 promoted cancer stemness by cancer stemness markers. (**a**) Spheroid cell formation was increased in NDRG1 overexpressed in HNC cells. After transfection of NDRG1-FL (FL) or the vector (Vt) plasmid into HNC cells (OECM1, FaDu), the effect of tumorsphere number and size was determined by microscope. (**b**) Overexpression of NDRG1-induced radioresistant ability in HNC cells, as determined by clonogenic survival assay. The error bars shown in the relevant figures indicate the standard deviation of the quantification results in the experiments. (*** *p* < 0.001; ANOVA test). (**c**) NDRG1 protected cisplatin-induced cell death, as determined by the MTS assay. (**d**,**e**) OECM1 and FaDu cells were transfected with NDRG1-FL or vector control and cells were collected for RT-qPCR analysis (**d**) or the cellular proteins were extracted and subjected to Western blot analysis and determination of protein expressions by antibodies (**e**). The GAPDH was used as an internal control. All the experiments were performed three times independently and typical results were shown. The error bars shown in the relevant figures indicate the standard deviation of the quantification results in all experiments. (* *p* < 0.05; ** *p* < 0.01; *** *p* < 0.001; n.s., no significance; *t*-test).

**Figure 3 cells-11-01581-f003:**
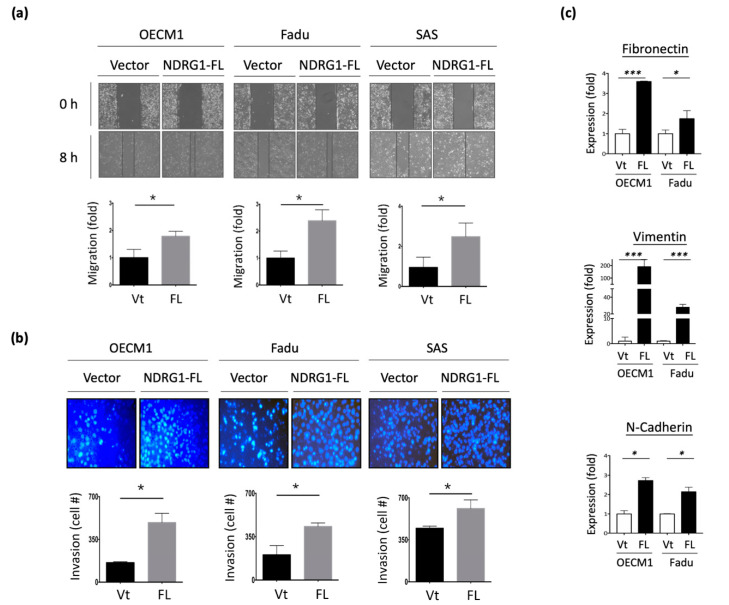
NDRG1 promoted cell motility via ECM mechanism. (**a**) NDRG1 facilitated cell migration in HNC cells. After seeding the NDRG1 overexpression (FL) or vector control (Vt) transfected cells into ibidi culture inserts for 8 h, cells were subjected to wound healing migration analysis. (**b**) NDRG1 increased cell invasion in HNC cells. After seeding the NDRG1 overexpression (FL) or vector control (Vt) transfected cells into Matrigel-coated membranes for 30 h, the cells were subjected to a Matrigel invasion assay. (**c**,**d**) OECM1 and FaDu cells were transfected with NDRG1-FL or vector control and cells were collected for RT-qPCR analysis (**c**) or the cellular proteins were extracted and subjected to Western blot analysis and determination of protein expressions by antibodies (**d**). The GAPDH was used as an internal control. All the experiments were performed three times independently and typical results were shown. The error bars shown in the relevant figures indicate the standard deviation of the quantification results in all experiments. (* *p* < 0.05; ** *p* < 0.01; *** *p* < 0.001; *t*-test).

**Figure 4 cells-11-01581-f004:**
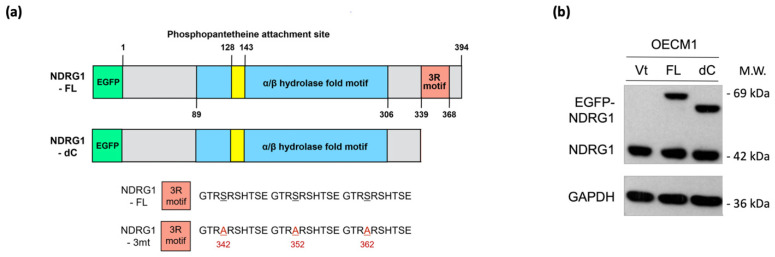
NDRG1-3R motif has minimal effect on cancer stemness. (**a**) Illustration of various NDRG1 truncated forms of EGFP tagged plasmid constructs (NDRG1-FL and NDRG1-dC). NDRG1-FL is composed of 394 amino acids; NDRG1-dC is composed of 337 amino acids, which lost the 3R motif. The mutations of GSK3b phosphorylation site of NDRG1 (NDRG1-3mt) are located at the 3R motif, which was made by triple serine-to-alanine substitutions (S342A, and S352A and S362A). (**b**) Confirmation of the various NDRG1 truncated forms’ expressions by Western blot analysis. After transfection of the NDRG1-FL, NDRG1-dC, or the vector plasmid into HNC cells, cellular proteins were extracted and subjected to Western blot analysis for NDRG1 expression. (**c**) Deletion of the C-terminal domain has no influence on NDRG1-induced cancer stemness. After transfection of the NDRG1-FL, NDRG1-dC, or the vector plasmid into HNC cells, the effect of NDRG1 on stem-like property was determined by spheroid cell formation. (**d**) Deletion of the C-terminal domain has no significant influence on NDRG1-induced radioresistance compared to NDRG1-FL, as determined by clonogenic survival assay in OECM1 cells. The error bars shown in the relevant figures indicate the standard deviation of the quantification results in the experiments. (* *p* < 0.05; n.s., no significance; ANOVA test). (**e**) After transfection of the NDRG1-FL, NDRG1-dC, or the vector plasmid into OECM1 cells, RNA was extracted and subjected to RT-qPCR analysis for cancer stemness markers. All the experiments were performed three times independently and typical results were shown. The error bars shown in the relevant figures indicate the standard deviation of the quantification results in all experiments. (* *p* < 0.05; ** *p* < 0.01; *** *p* < 0.001; n.s., no significance; *t*-test).

**Figure 5 cells-11-01581-f005:**
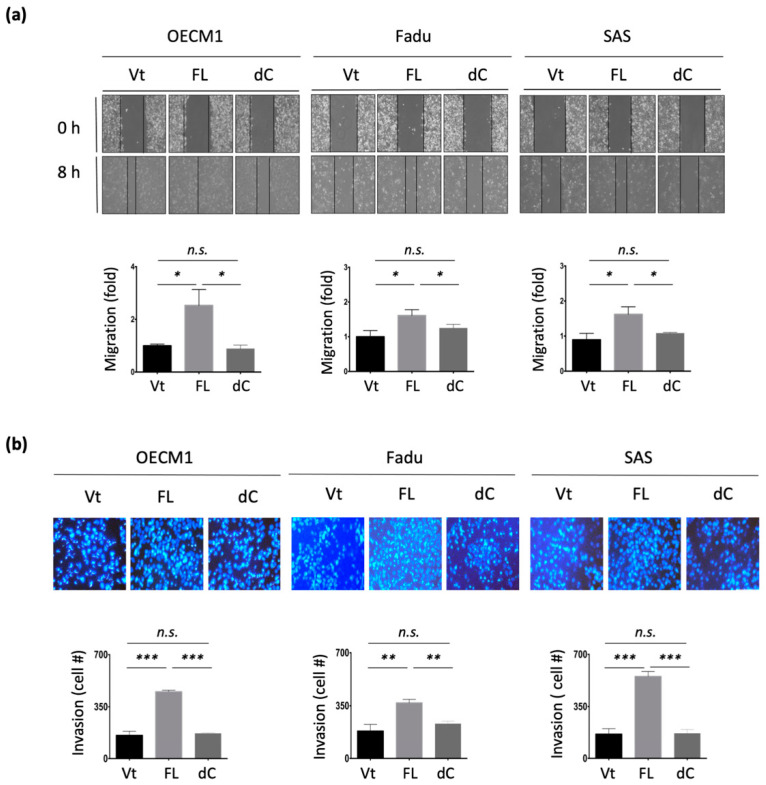
NDRG1-3R motif is critical for cellular motility. (**a**) Deletion of the C-terminal domain significantly attenuated NDRG1-induced cell migration. After transfection of the NDRG1-FL, NDRG1-dC, or the vector plasmid into HNC cells, the cells were subjected to wound healing migration analysis. (**b**) Deletion of the C-terminal domain significantly inhibited NDRG1-induced cell invasion. After transfection of the NDRG1-FL, NDRG1-dC, or the vector plasmid into HNC cells, the cells were subjected to Matrigel invasion assay. (**c**) Deletion of the C-terminal domain suppressed NDRG1-induced ECM-associated molecules. After transfection of the NDRG1-FL, NDRG1-dC, or the vector plasmids into HNC cells, cellular proteins were extracted and subjected to Western blot analysis. The GAPDH was used as an internal control. All the experiments were performed three times independently and typical results were shown. The error bars shown in the relevant figures indicate the standard deviation of the quantification results. (* *p* < 0.05; ** *p* < 0.01; *** *p* < 0.001; n.s., no significance; *t*-test).

**Figure 6 cells-11-01581-f006:**
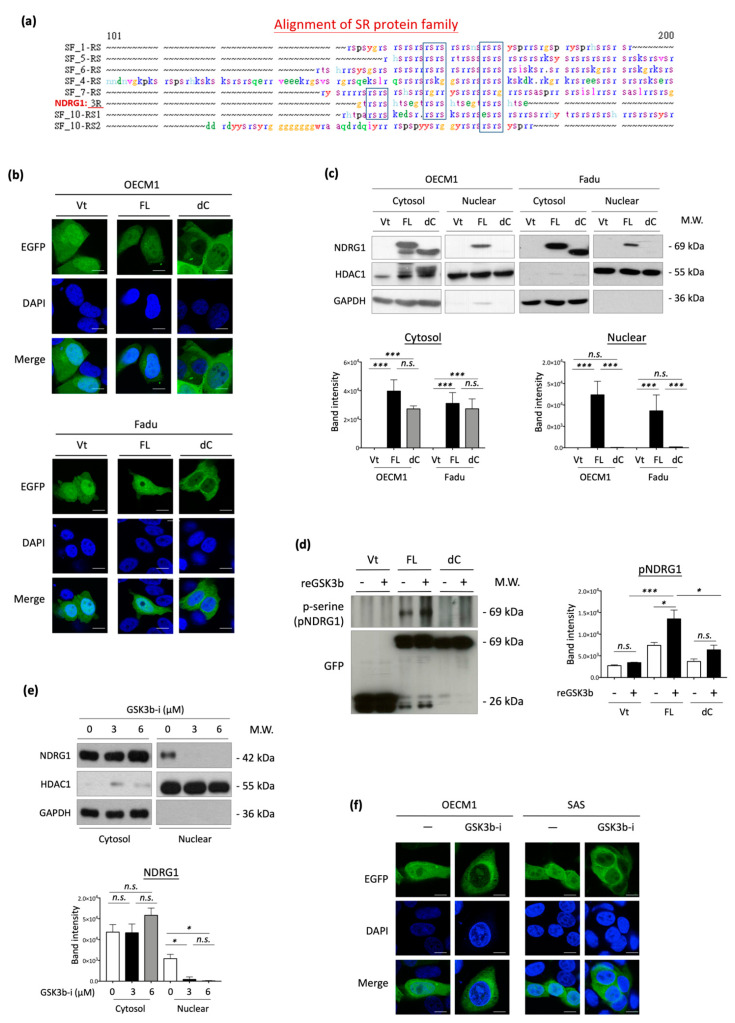
The 3R-motif of NDRG1 is actioned for nuclear translocation and phosphorylation. (**a**) Sequence homology of NDRG1-3R motif with SR protein family. The sequences of 3R motif and SR family proteins were aligned using Pileup program in Chang Gung University. The conserved amino acids are blocked in the indicated boxes. (**b**) Depletion of NDRG-3R motif significantly inhibited nuclear translocation in OECM1and FaDu cells. After transfection of NDRG1-FL, NDRG1-dC, or the vector plasmid into HNC cells, the cells were subjected to immunofluorescent and confocal microscopic analysis. Because the EGFP protein sequence was fused in-frame in the NDRG1 expression plasmids, the localization of EGFP indicated the specific localization of NDRG1. Blue staining with DAPI dye was used to indicate nuclei, scale bar 10 µm. (**c**) Depletion of 3R motif significantly inhibited nuclear translocation. After transfection of NDRG1-FL, NDRG1-dC, or the vector plasmid into HNC cells, the subcellular compartment proteins were isolated and separated to nuclear and cytosolic fractions. The NDRG1 expression level in various subcellular fractions was determined by Western blot analysis. The HDAC1 and GAPDH were used as internal controls for nuclear and cytosolic compartment proteins. (**d**) GSK3b phosphorylated NDRG1 at the serine residues of its C-terminal domain. After transfection of the NDRG1-FL or NDRG1-dC expression plasmids into OECM1 cells, the cellular proteins were extracted and subjected to an in vitro phosphorylation assay with recombinant GSK3b kinase. These NDRG1 proteins were then pulled down with anti-GFP antibody, and their phosphorylation status was revealed by immunoblotting with phosphoserine-specific antibody. Expression of the EGFP protein in each transfected sample was used as an internal control. (**e**) Treatment of GSK3b inhibitor suppressed nuclear translocation of NDRG1 in HNC cells. The OECM1 cells were treated with GSK3b-specific inhibitor (SB21673) at various doses as indicated for 24 h. The subcellular compartment proteins were isolated. The endogenous NDRG1 expression level in various subcellular fractions was determined by Western blot analysis. The HDAC1 and GAPDH were used as internal controls for nuclear and cytosolic compartment proteins, respectively. (**f**) Treatment of GSK3b inhibitor attenuated nuclear translocation of NDRG1 in HNC cells. After transfection of the NDRG1-FL or NDRG1-dC expression plasmids into HNC cells, transfected cells were treated with GSK3b-specific inhibitor (SB21673) at 3 μM for 24 h. These cells were then subjected to immunofluorescence analysis and confocal microscopy, as described in the method section, scale bar 10 µm. The error bars shown in the relevant figures indicate the standard deviation of the quantification results. (* *p* < 0.05; *** *p* < 0.001; n.s., no significance; *t*-test).

**Figure 7 cells-11-01581-f007:**
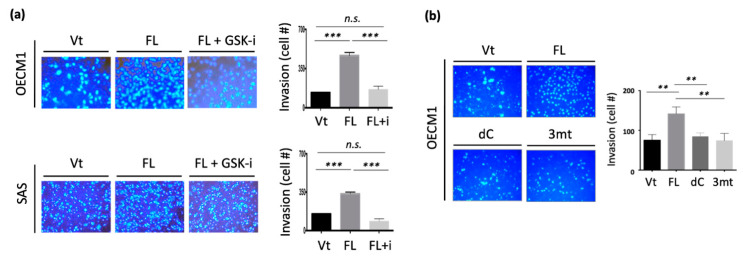
The phosphorylation of 3R-motif by GSK3b is critical for cell motility in HNC cells. (**a**) Dephosphorylation of NDRG1 by GSK3b inhibitor reversed the effect of NDRG1-induced cell invasion. The HNC cells (OECM1, SAS) were transfected with NDRG1-FL or the vector plasmids, with or without the addition of GSK3b inhibitors (SB21673, 3 μM) for 24 h. These cells were then subjected to Matrigel invasion assay. (**b**) Mutation of GSK3b phosphorylation sites in NDRG1-3R motif reversed the effect of NDRG1-induced cell invasion. After transfection of various NDRG1 truncated plasmids (-FL, -dC, -3mt) into OECM1 cells for 30 h, these cells were subject to Matrigel invasion assay. All the experiments were performed three times independently and typical results were shown. The error bars shown in the relevant figures indicate the standard deviation of the quantification results in all experiments. (** *p* < 0.01; *** *p* < 0.001; n.s., no significance; *t*-test).

**Figure 8 cells-11-01581-f008:**
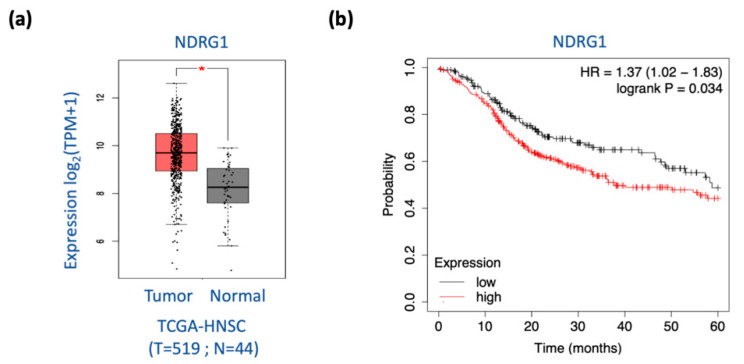
High level of NDRG1 is significantly associated with poor survival in HNC patients. (**a**) In HNC patients, NDRG1 is overexpressed in the tumors compared to the normal tissues from TCGA-HNSC datasets generated by GEPIA2 website. The relative expression levels were shown. (**b**) Higher level of NDRG1 expression is associated with poor prognosis in HNC patients, as determined by Kaplan–Meier survival analysis via KM-Plotter online tool from the HNC dataset (N = 500). * *p* < 0.05; *t*-test.

## Data Availability

The data of HNC patients used in this study are from the publicly accessible GEO and KM-Plotter online tool databases, which offer anonymous data.

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
