# Peer review of "Multifaceted and Intricate Oncogenic Mechanisms of NDRG1 in Head and Neck Cancer Depend on Its C-Terminal 3R-Motif"

_cells, 2022, doi:10.3390/cells11091581_

Round 1
Reviewer 1 Report
The authors study the role of NDRG1 in regulating HNC progression. They reveal that cytoplasmic NDRG1 confers cancer stemness and therapeutic resistance, while NDRG1 in nucleus likely facilitate motility. Especially NDRG1 may translocate into the nucleus via phosphorylation at its 3R-motif to facilitate motility. The data elucidate the multifaceted and intricate functions of NDRG1 in HNC. Overall, the data is interesting, however the paper provides for descriptive observations wherein correlative analysis was used to derive at conclusions in the absence of novel and rigorous mechanistic studies. There are some issues that need to be addressed before the manuscript is acceptable for publication.
- The whole ideas seem missing some important evidences in vivo, e.g. tumor growth in animal models; analysis the localization and expression of NDRG1 in tumor tissue section from HNC patients. Is there any correlation between NDRG1 expression and chemoresistant of HNC patients?
- Although NDRG1 is involved in several cancer progressions, the authors should provide stronger rationales to study the function of NDRG1 in HNC, because hundreds or thousands of genes are associated with cancer progression. Is there anything special about NDRG1?
- The authors should make a statement regarding their rationale for selecting certain cell lines for their studies. How about endogenous NDRG1? Are they mainly located in nuclear or cytoplasm?
- How were the 1109 DEGs identified? By analyzing the differential transcriptomes from OECM1 cells, or all three cell lines? Please clarify.
- In Fig.1d, GSEA analysis suggested that NDRG1 is associated with EMT, and NDRG1 facilitated the expression of EMT-related makers in Fig 3, such as N-cadherin and Vimentin, however, the immunofluorescence analysis in Fig.6b didn’t supported morphological change of EMT when NDRG1 is overexpressed. Please clarify.
- Please check the use of the correct statistical test. For example, Fig. 4d should be tested by ANOVA (multiple group comparisons) and not t-test.
- Please perform statistical analysis for qPCR analysis and marked whether there is statistical significance in Fig 2d, 3c, 3e.
- Please provide scale bars in Fig 6b and f.
- Please marker the molecular weight in the results of western blot analysis (e. g., Fig1a; Fig2e; Fig3d; Fig4b; Fig5c, and Fig. 6d and e), and also indicate endogenous and exogenous NDRG1.
- Why were the authors focused on 3R motif of NDRG1? Why not other domain of NDRG1, such as α/β hydrolase fold motif? The authors should make a stronger rationale.
- In Fig.6b, the immunofluorescence analysis indicated that the protein of NDRG1 distributed mainly in the nuclear, while the expression of NDRG1 seemed to be much higher in cytoplasm in Fig. 6c comparing with NDRG1 expression in nuclear. Please clarify.
- The authors have demonstrated that overexpression of NDRG1 promoted the HNC progression, however, have the authors observed the similar phenotype change when NDRG1 was blocked or knocked down?
- The authors reveal that cytoplasmic NDRG1 confers cancer stemness and therapeutic resistance, while NDRG1 in nucleus likely facilitate motility. How do the authors explain this phenomenon? Please list some recent publications on NDRG1 to compare the difference of physiological function of NDRG1 in cytoplasm and in nuclear.
Author Response
To Reviewer and Editor:
We thank your comments that help us to make this manuscript more significant. All the comments were well taken and amended in the revised manuscript, using the “track marked” function to make changes. We have made a point-by-point response to your comments as below. Please see the attachment. Thank you very much!

Reviewer 2 Report
In You & Chang et al., the authors discussed the role of NDRG1 in head and neck cancer. Using Microarray, the authors showed that NDRG1 induces various oncogenic-related processes including growth, migration and stress responses. The authors used mostly two cell lines (namely, OECM1 and FaDU) in their subsequent experiments. The study demonstrated that NDRG1 over-expression resulted in increased resistance to irradiation. Moreover, the authors demonstrated an increased expression of stemness markers (e.g. CD44, Twist1, BMI1, NES) upon the over-expression of NDRG1. Furthermore, the authors showed that the over-expression of NDRG1 resulted in increased cell motility and upregulation of ECM genes (i.e Fibronectin, Vimentin, N-Cadherin and SNAIL). The authors then focused on the 3R motif of the protein and over-expressed a version of the protein that lacks the motif. They proved that the motif was not essential for the role of protein in stemness induction or for irradiation resistance. However, they demonstrated that it is required for the cell invasion capability of the malignant cells. They also exhibited its role in the nuclear translocation of the protein. Finally, the authors showed that NDRG1 is significantly unregulated in Tumor samples and correlated with lower survival rate in head and neck cancer using the TCGA cohort.
Overall, the study is interesting especially in studying the role of the 3R motif in terms of head and neck cancer. However I have the following concerns regarding the study:
1- The methods for the bioinformatic analysis of the microarray has to be discussed in more details to assess the results reported.
2- The sequence of the qPCR primers used for CD44 demonstrated non specificity for the primer. Hence results based on this primer should not be trusted.
3- The primers used for Vimentin does not show any amplification in in silico PCR which is very alarming. We can not trust the results from the Vimentin primers.
4- The sequence of the qPCR primers used for SNAI1 demonstrated non specificity for the primer. Hence results based on this primer should not be trusted.
5- Figure 3c and Figure 3d lack statistical testing.
6- Bar charts for density of the gel plot is required for Figure 3d.
7- Figure 2d and Figure 2e lack statistical testing and density plot for the gel plot as well.
8- The title and discussion indicate only head and neck cancer. However, OECM1 cell line is an oral cancer cell line. The authors need to discuss that and re-write parts of the manuscript to indicate that.
9- It is crucial that the authors discuss why their result contradict the findings of Lee et al 2014 (Cancer Letters).
Author Response

(The authors gave the same response as above.)

Reviewer 3 Report
You and colleagues presented an article on N-Myc downstream-regulated 1 (NDRG1) investigating its intricate oncogenic mechanisms in Head and Neck Cancer (HNC). Through in vitro studies, the authors highlighted that NDRG1 is involved mainly in stress response, including radio-chemoresistance and cell motility. In particular, they demonstrated that C-terminal 3R-motif and its GSK3b-induced phosphorylation are essential for nuclear import and modulation of genes involved in cancer cell migration.
Moreover, they conducted an in silico analysis demonstrating that NDRG1 was overexpressed in HNC patients and high levels of this protein were correlated with poor prognosis. The paper is well written and the used methodologies are appropriated for the aim of the study. Overall, the obtained results encourage further investigations on NDRG1 as potential prognostic indicator or therapeutic target for HNC. However, I have some major comments as follows:
- In 2.2 section of Material and Method, please add further information about “point mutagenesis”.
- In 2.3 section, please better explain the used samples/cell lines for the transcriptomic analysis.
- In section 2.7 add qPCR amplification condition.
- In section 2.5 please supply additional information on MTS assay.
- In Material and Method section, the authors reported that three HNC cell lines were used to perform in vitro studies. However, in most of figures they described only two cell lines (e.g. Figures 1d, 2a, 2b, 2d, 2e, 3c, 3d, 4, 5c 6b, 6c, 6f, 7a, 7b). The missing results should be reported or discussed in the main text if no significant.
- The statistical significance is not reported for all histograms (e.g. Figures 2d, 3c, 4e, 5c, 6c, 6d). Add the significance and discuss the results in the main text.
- In some figures is not clear the HNC cell line used for the in vitro studies (e.g. Figures 4d, 4e, 6d, 6e). Clarify and comment the results in the main text.
- In Figure 1a NDRG1 must be change with OECM1 to indicate the cell line.
- In line 287 the authors reported that NDRG1 overexpression resulted in higher drug resistance at concentration of 5 μg/mL in Fadu cells. This result does not match to Figure 2c.
- In line 291, the authors reported that “Although there were various levels in the two HNC cell lines, these molecules were generally elevated in response to NDRG1 transduction”. The result described in the main text does not match to Figure 2d. The expression levels of the analyzed cellular proteins are different between the two considered cell lines. Please add the statistical significance.
- The Western blot of all analyzed cellular proteins should be added and described in the main text (e.g. Figures 2e, 3d).
- Figure 9 could be used as graphical abstract.
Author Response

(The authors gave the same response as above.)

Round 2
Reviewer 2 Report
In this round of revision, You & Chang et al. have improved the manuscript. I appreciate the correction of Vimentin primer sequence. Statistical tests were performed for all quantitative analyses. I have the following minor concerns which I believe should be addressed before considering the manuscript for publication:
1- I recommend using Primer-Blast instead of just using Blast to determine Primer specificity (Even if the primers were used in previous investigations). Many published primers are unfortunately not specific and suitable for qPCR. Given that most of your primers are specific and good, I recommend removing CD44 and SNAI1 from the results. According to in-silico PCR (Primer-Blast) these primers have potential off target amplification that could impair proper interpretation.
2- In the methods section such statements are discouraged, "The transcriptomic and bioinformatic analyses were performed similarly as previously described". Please provide brief details after this sentence. (i.e What software was used; What were the ceiling values; Which normalization method was used; DGE measurement and Thresholds).
3- I appreciate adding 'Lee et al' to the discussion. It will be great to add a sentence about the cellular or experimental conditions in that manuscripts which could potentially lead to this contrasting finding. It gives more assurance to your study.
Author Response
To Reviewer 2:
We thank your comments that help us to make this manuscript more significant. All the comments were well taken and amended in the revised manuscript, using the “track marked” function to make changes. We have made a point-by-point response to your comments as below. Please see the attachment. Thank you very much!

Reviewer 3 Report
Dear Authors,
Thank you for your reply to my comments. No further changes are required.
Best regards.
Author Response
To Reviewer 3:
We thank your comments that help us to make this manuscript more significant. Please see the attachment. Thank you very much!
Best regards,
Ann-Joy Cheng,
Chang Gung University
